# Demographic history and population structure of *Phlebotomus argentipes* (Diptera: Psychodidae) complex, the leishmaniasis vector in Sri Lanka

W. M. M. Wedage[1], Iresha N. Harischandra[2,3], O. V. D. S. J. Weerasena[4],
S. A. S. C. Senanayake[5], B. G. D. N. K. De Silva[1,6]*

**1** Center for Biotechnology, Department of Zoology, Faculty of Applied Sciences, University of Sri Jayewardenepura, Nugegoda, Sri Lanka, **2** Genetics and Molecular Biology Unit, Faculty of Applied Sciences, University of Sri Jayewardenepura, Nugegoda, Sri Lanka, **3** Vidya Sethu Foundation, Thalangama North, Battaramulla, Sri Lanka, **4** Institute of Biochemistry, Molecular Biology and Biotechnology (IBMBB), University of Colombo, Kumaratunga Munidasa Mawatha, Colombo, Sri Lanka, **5** Department of Parasitology, Faculty of Medicine, University of Colombo, Colombo, Sri Lanka, **6** Sri Lanka Institute of Biotechnology (SLIBTEC), Pitipana, Homagama, Sri Lanka

* nissanka@sci.sjp.ac.lk

## Abstract

*Phlebotomus argentipes* sensu lato Annandale & Brunetti, 1908 is the primary vector of *Leishmania donovani* MON 37, the causative agent of cutaneous leishmaniasis (CL) in Sri Lanka. Effective vector control is essential for managing leishmaniasis. Although numerous taxonomic studies have been conducted on *P. argentipes* s.l., population genetics remains insufficiently explored. This study investigated the demographic history and population genetic structure of *P. argentipes* s.l. in Sri Lanka using sequence data obtained from the earlier investigation of two mitochondrial markers, Cytochrome c oxidase subunit I (*COI*) and NADH dehydrogenase subunit 4 (*ND4*). For the genetic analysis, 159 individuals from five leishmaniasis endemic sites were examined. In addition to the individual analyses of *COI* and *ND4* genes, a concatenated dataset combining both mitochondrial fragments was constructed to evaluate overall genetic structure and demographic history. The population structure and demographic history of *P. argentipes* s.l. were assessed using $F_{ST}$ estimates, AMOVA, structure analysis, Mantel test, PCoA, Bayesian inference and coalescent analysis. The highest $F_{ST}$ value was 0.0271, indicating low genetic differentiation, with over 98% variation occurring within populations. Mantel tests showed weak, non-significant correlations between genetic and geographic distance, indicating no evidence of isolation by distance, suggesting potential gene flow and no distinct clustering within the Sri Lankan *P. argentipes* s.l. population. Negative and significant neutrality statistics, together with unimodal mismatch distributions, support historical population expansion, further corroborated by Bayesian skyline plots indicating two distinct demographic events, an ancient expansion around 50,000 years ago (*COI*)

**Data availability statement:** All relevant data are within the manuscript and its Supporting Information files.

**Funding:** This research was funded by the University Research Grant, University of Sri Jayewardenepura, grant number ASP/01/RE/SCI/2017/53, and the Centre for Biotechnology, Department of Zoology, Faculty of Applied Sciences, University of Sri Jayewardenepura, Sri Lanka. Snr. Prof. B.G.D.N.K. De Silva received the funding. The funders had no role in study design, data collection, analysis, the decision to publish, or manuscript preparation.

**Competing interests:** The authors have declared that no competing interests exist.

and a more recent one approximately 10,000–13,000 years ago (*ND4*). Additionally, the mismatch distribution analyses revealed a multimodal expansion pattern at the Medirigiriya and Hambantota sites, which are hot spots for leishmaniasis in Sri Lanka. The present study demonstrates a demographic expansion and genetic homogeneity of *P. argentipes* s.l. populations in Sri Lanka, supporting the species' ability to colonize new areas and possibly enhance leishmaniasis transmission. This connectivity may facilitate the spread of adaptive traits such as insecticide resistance, even in the absence of local selection pressure, posing a potential challenge for future vector control efforts in Sri Lanka.

## Introduction

Cutaneous leishmaniasis poses a significant public health challenge in many tropical and subtropical regions, including Sri Lanka. *P. argentipes* s.l. Annandale & Brunetti, 1908 [1,2] is the primary vector, transmitting the *Leishmania donovani* MON 37 parasite, the causative agent in the country [3]. The increasing cases of leishmaniasis in Sri Lanka have been linked to the expansion of *P. argentipes* s.l. populations, which aligns with their adaptation and ecological competence under diverse environmental conditions [4–6]. A thorough understanding of vector populations, their structure and dynamics is essential for effective disease control, as population genetic studies reveal patterns of gene flow, dispersal, and adaptation [7].

In recent years, the endemic pattern of leishmaniasis in Sri Lanka appears to have shifted significantly, possibly due to changes in the distribution and behavior of *P. argentipes* s.l. [6,8,9]. Initially, leishmaniasis, which had been confined to Sri Lanka's dry zone, is now emerging in districts bordering the country's wet zones [8,10–12]. The key factor facilitating the rapid spread of the disease is the proliferation of the infected *P. argentipes* s.l. vector throughout the area [13,14]. Furthermore, it has been suggested that population expansion allows sandfly vectors to adapt to diverse environmental settings and anthropogenic influences. This adaptation can lead to increased dispersal into previously unaffected areas. Although sandflies typically disperse only short distances (a few hundred meters), population expansion over multiple generations, aided by adaptation to diverse environments and anthropogenic influences may have gradually enabled a range expansion [15]. Occasional long-distance dispersal, possibly driven by passive transport (e.g., wind currents or human activities), could further enhance gene flow between populations. However, such expansion might also involve founder effects or genetic bottlenecks, particularly when colonization occurs with only a limited number of individuals [16].

Numerous studies on the *P. argentipes* complex have contributed to understanding its taxonomy [2,14,17,18], but studies on population genetics within Sri Lanka remain limited. Previous studies have examined the genetic diversity of *P. argentipes* s.l. populations in Sri Lanka using mitochondrial markers, reporting relatively low variability across regions [19,20]. However, the persistence of central nucleation within radiating haplotype networks observed in these populations [20] suggested retention

of ancestral haplotypes, which can contribute to overall genetic diversity. Pathirage et al. [19] indicate that non-targeted vector control efforts may have unintentionally contributed to the development of insecticide resistance and changes in the genetic makeup of *P. argentipes* s.l. populations. Insecticide use can exert selective pressure on *P. argentipes* s.l., promoting the spread of resistance alleles such as *kdr*. These selective pressures can influence genetic structure and potentially lead to population differentiation [19]. Additionally, factors such as the movement of infected humans, migrant laborers, and domestic animals such as dogs increasingly recognized as potential reservoirs for *L. donovani* may also influence the recent (observed) population expansion of *P. argentipes* s.l., the vector [21]. Nevertheless, population structure, gene flow patterns, and demographic history are still largely unexplored, emphasizing the need for a more comprehensive population genetic analysis of *P. argentipes* s.l.

The present study addresses this knowledge gap by analyzing genetic data from extant populations in Sri Lanka to infer their demographic history and population structure. By integrating historical, geographical, and behavioral factors, this analysis aims to reveal the complex drivers of genetic diversity within *P. argentipes* s.l. populations in Sri Lanka. Understanding gene flow and population expansion patterns among modern representatives of *P. argentipes* s.l. is essential for effective leishmaniasis control, as it remains a major health concern in Sri Lanka.

## Materials and methods

### Data set

The current study was conducted using the same mitochondrial sequence dataset (*COI* and *ND4*) published previously [20], with the aim of applying additional population genetic analyses to explore gene flow, population structure, and demographic history across *P. argentipes* s.l. populations in Sri Lanka.

### Original sample collection and sequencing

The original dataset comprises *P. argentipes* s.l. individuals collected from five geographical locations in Sri Lanka: Anuradhapura (6.313056°N, 81.224167°E), Balangoda (6.671942°N, 81.023608°E), Mirigama (7.258331°N, 80.150831°E), Medirigiriya (8.143417°N, 81.039300°E), and Hambantota (6.313056°N, 81.224167°E) between March 2018 and March 2020 [20] (S1 Table). The collection of sandfly individuals was carried out using cattle-baited net traps, light traps, sticky traps and manual collections. DNA extraction, PCR amplification, and sequencing of the *COI* and *ND4* genes were performed following standard protocols [20–23].

### Population structure analysis

Two gene fragments, *COI*, *ND4,* and the concatenated sequences, were aligned and trimmed using ClustalW in MEGA 5.2 [24], resulting in final alignment lengths of 455 bp, 598 bp and 1053 bp, respectively. Sequences were validated via BLAST (NCBI) and deposited in GenBank [20]. Pairwise $F_{ST}$ values for three data sets were computed using Arlequin v.3.5 software [25]. The relationship between genetic distance and geographic distance was analyzed through the Mantel test [26]. Geographical distances were calculated using ArcGIS [27] and plotted against pairwise linearized $F_{ST}/ (1- F_{ST})$ to assess the correlation between geographic and genetic distances. Pairwise $F_{ST}$ and corresponding distances were computed, using 1000 permutations, with a significant level of 0.05 for all markers.

The population structure results were validated using $F_{ST}$ values (Fixation Indices) through the AMOVA (Analysis of Molecular Variance) test implemented in Arlequin v.3.5 software [28].

Principal coordinates analysis (PCoA) based on the genetic distance matrix was performed using the simple matching coefficient and the decenter and eigenvector matrices in GenAlEx with 1000 random permutations (GenAlEx v. 6.5) as an Excel add-in tool kit [29]. The mismatch distribution analysis was carried out to find evidence of past demographic inference [30]. Additionally, this analysis was used to estimate the expansion parameters $\theta^0$, $\theta^1$ and Tau ($\tau$), the time of

the expansion measured in units of mutational time ($\tau = 2\,\mu t$; $t$ is the time in generations). Raggedness index r [31], whose significance was tested through 1000 replicates, was calculated to assess the significance of the inferred expansion. Neutrality test values, Tajima's D and Fu's *Fs*, were calculated based on segregation sites to support for indications of a past population expansion. All these analyses were performed using DnaSP and Arlequin v.3.5 [25].

## Bayesian inference and coalescent analysis

The partition finder v.1.1.1 was used to identify the best-fitting Bayesian Information Criterion (BIC) model for the two datasets. Coalescent Bayesian Skyline Plots (BSP) were generated for the current study based on two datasets of *COI* and *ND4* genes to explore potential demographic changes further using BEAUti2 [32], BEAST2 [32], and Tracer 1.6 [33]. Based on the mutation rates in *Drosophila* mtDNA, the substitution rate for BEAUti2 was set to $6.2 \times 10^{-8}$ per site per generation [34]. The gamma category count was set to 4; the shape parameter to 1.0; the proportion invariant to 0.5; and the evolutionary model set to HKY. BEAUti2 was also configured to estimate all aforementioned parameters during Bayesian analysis. A strict clock model was applied, with a rate of 1.0. The tree model was set to Coalescent Bayesian Skyline with Random Tree prior in the Priors menu. The chain length was set to 30 million MCMCs, with a burn-in of 3 million, and convergence was assessed visually using Tracer 1.6. Convergence was deemed to be reached when the trace plot displayed mean and variance patterns consistent with stationarity.

## Results

### Variation in the mitochondrial COI and ND4 genes

The nucleotide composition within the *COI* and *ND4* sequences exhibited a high degree of similarity when considering the average values across the multiple sequence alignment (Table 1). According to the concatenated data set for both mtDNA

**Table 1. Molecular diversity indices and neutrality test, based on *COI, ND4* and concatenated sequences obtained from five studied *Phlebotomus argentipes* s.l. populations.**

| Sequence alignment | Population ID | Number of sequences used | π | h | Hd | Fu's Fs | Tajima's D | Fu and Li's D | Fu and Li's F | Reference |
|---|---|---|---|---|---|---|---|---|---|---|
| Cytochrome Oxidase I | ANU | 25 | 0.00406 | 14 | 0.81 | −10.42 | −2.2763* | −3.57677* | −3.71736* | Wedage et al., 2023 [20] |
| | BAL | 31 | 0.00348 | 14 | 0.8 | −10.22 | −2.2974* | −3.9784* | −4.04679* | |
| | MIR | 28 | 0.00232 | 9 | 0.55 | −5.273 | −2.2923* | −3.68088* | −3.80693* | |
| | MED | 24 | 0.00354 | 11 | 0.75 | −6.463 | −2.2238* | −3.29557* | −3.46897* | |
| | HAM | 40 | 0.00417 | 18 | 0.75 | −13.82 | −2.4776* | −4.07009* | −4.1793* | |
| NADH dehydrogenase subunit 4 *(ND4)* | ANU | 27 | 0.00453 | 17 | 0.94 | −11.87 | −1.702 | −1.98299 | −2.22239 | |
| | BAL | 29 | 0.00464 | 18 | 0.93 | −12.72 | −1.8705* | −2.90004* | −3.09982* | |
| | MIR | 21 | 0.00483 | 12 | 0.91 | −5.133 | −2.0006* | −2.83431* | −3.06489* | |
| | MED | 25 | 0.01055 | 21 | 0.98 | −12.32 | −2.253* | −3.45401* | −3.61152* | |
| | HAM | 31 | 0.00517 | 18 | 0.83 | −10.83 | −2.4642* | −4.75866* | −4.72714* | |
| Concatenated data set | ANU | 25 | 0.00432 | 22 | 0.98 | −18.671 | −2.1259* | −2.95229 | −2.9132* | This study |
| | BAL | 28 | 0.00409 | 25 | 0.98 | −24.557 | −2.1662* | −3.46003 | −3.30167* | |
| | MIR | 17 | 0.00286 | 12 | 0.919 | −6.395 | −1.9407 | −2.4225* | −2.41394* | |
| | MED | 21 | 0.00731 | 18 | 0.981 | −8.050 | −2.38793 | −3.45207* | −3.34938* | |
| | HAM | 30 | 0.00481 | 22 | 0.933 | −13.093 | −2.60505 | −4.39994 | −4.14913 | |

Abbreviations: ANU- Anuradhapura, BAL- Balangoda, MIR- Mirigama, MED- Medirigiriya, HAM- Hambantota, π- Nucleotide diversity per site, k- Average number of nucleotide differences, h- Number of haplotypes, Hd- Haplotype diversity. * Significant at $0.01 < p < 0.05$, while other values are non-significant ($p > 0.10$) for neutrality tests.

 

fragments, there was a significantly greater haplotype diversity and nucleotide diversity in the Medirigiriya population; however, the highest haplotype count (h = 25) was observed in the Balangoda population (Table 1).

## The population genetic structure

Based on alignments of the *ND4* and *COI* sequences, the pairwise $F_{ST}$ within and between populations in the study are shown in Table 2. Population pairwise $F_{ST}$ values were highest for *ND4* and *COI* sequence alignments between the Hambantota, Medirigiriya populations and the Mirigama, Balangoda populations respectively (Table 2). The lowest $F_{ST}$ for *COI* occurred between the Balangoda and Anuradhapura sites. The lowest $F_{ST}$ for *ND4* was between the Balangoda and Mirigama sites (Table 2).

According to the concatenated pairwise $F_{ST}$ matrix, Balangoda and Hambantota sites suggest there may be slight but significant differentiation with statistical support (p < 0.05) (Fig 1). Analysis of the inter-haplotype pairwise distance matrix based on concatenated alignment revealed variable levels of intra-population genetic divergence across the five sampling sites, with Medirigiriya and Anuradhapura showing the highest range of nucleotide differences, while Mirigama exhibited comparatively low divergence among haplotypes (S1 Fig).

The Mantel test revealed no correlation between genetic distance and geographic distance for the concatenated data set (Fig 2C) and *COI* (Fig 2A); although *ND4* sequences (Fig 2B) suggested a correlation, this appeared inconsistent with the overall signal of population expansion.

**Table 2. Pairwise $F_{ST}$ values for five *Phlebotomus argentipes* s.l. populations based on the *COI* and *ND4* genes.**

| Population | ANU | BAL | MIR | MED | HAM |
|---|---|---|---|---|---|
| ANU | | −0.00928 | −0.00052 | −0.00107 | −0.00309 |
| BAL | −0.00337 | | 0.00436 | 0.00404 | −0.00247 |
| MIR | 0.00318 | −0.00738 | | −0.00736 | −0.00153 |
| MED | −0.00613 | −0.00649 | 0.00597 | | 0.00327 |
| HAM | 0.01544 | 0.01184 | 0.00299 | **0.0271\*** | |

The value below the diagonal indicates the population pairwise $F_{ST}$ values for the *ND4* gene fragment. Values above the diagonal indicate the population pairwise $F_{ST}$ values for the *COI* gene fragment. * Significant value (p = 0.0478).

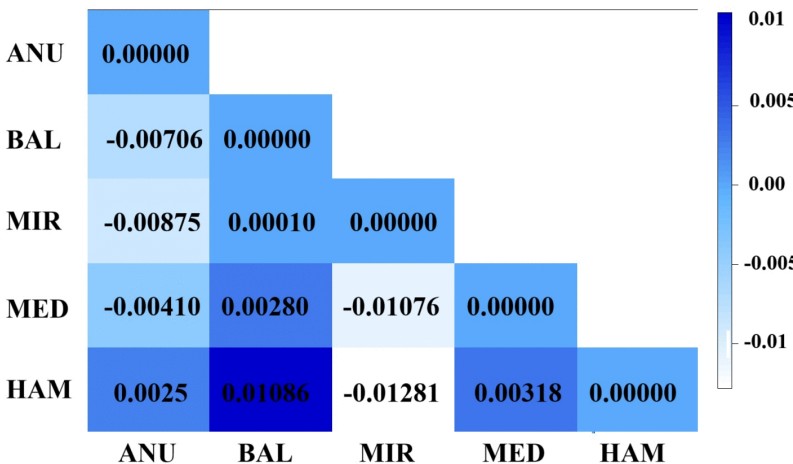

**Fig 1. Pairwise $F_{ST}$ matrix of five studied populations of Phlebotomus argentipes s.l. based on concatenated alignment.**

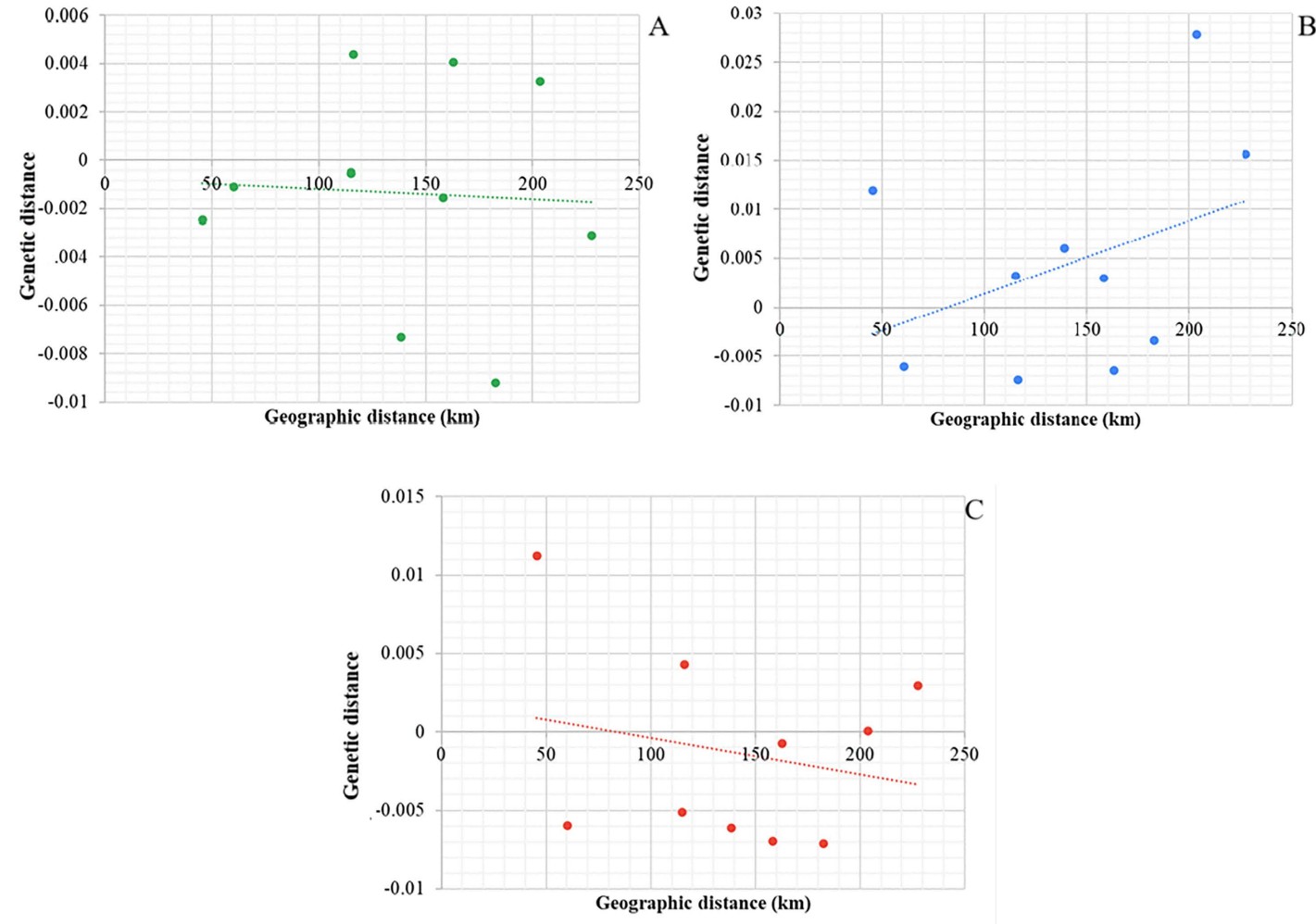

**Fig 2. The relationship between genetic distance expressed as linearized $F_{ST}/(1- F_{ST})$ and pairwise geographic distance (km) of the *Phlebotomus argentipes* s.l. field populations. (A)** *COI* sequences, **(B)** *ND4* sequences **(C)** concatenated sequences and trendline- linear regression.

For the *COI* dataset, the Mantel correlation coefficient was r = −0.038 (p = 0.48), for the *ND4* dataset, r = 0.040 (p = 0.52), and for the concatenated dataset, r = −0.043 (p = 0.43). None of the correlations were statistically significant, indicating no isolation by distance among the studied populations.

AMOVA revealed that almost all the total molecular variance (>99.6%) was attributed to differences among individuals within populations, while the proportion of variation among populations was negligible (<0.4%) for all datasets. For the *COI* fragment, the variance component among populations was −0.00019 (−0.05% of variation; $F_{ST}$ = −0.00052, p > 0.05). For *ND4*, the variance component among populations was 0.00662 (0.38% variation; $F_{ST}$ = −0.00375, p > 0.05). The concatenated dataset showed a variance component of −0.00204 (−0.08% of variation; $F_{ST}$ = −0.00082, p > 0.05). In all cases, the lack of statistical significance indicates an absence of substantial genetic differentiation among *P. argentipes* s.l. populations in Sri Lanka by supporting demographic expansion rather than pronounced population subdivision. (Table 3).

According to the pattern of genetic variation, the first principal coordinate explained 18.06% of the total variation, while the second component accounted for 8.77% in the *COI* dataset. Together, the first three axes explained 31.77% of the

**Table 3. Analyses of molecular variance (AMOVA) for testing the genetic subdivision of five *Phlebotomus argentipes* s.l. field populations in Sri Lanka using two mtDNA region sequences across geographic districts.**

| Data set | Source of variation | df | Sum of squares | Variance components | Percentage of variation | F- statistics |
|---|---|---|---|---|---|---|
| *COI* | Among populations | 4 | 1.442 | −0.00019 Va | −0.05 | $F_{ST}$=−0.00052 p>0.05 |
| | Within population | 143 | 52.342 | 0.36603 Vb | 100.05 | |
| *ND4* | Among groups | 4 | 7.733 | 0.00662 Va | 0.38 | $F_{ST}$=−0.00375 p>0.05 |
| | Within population | 128 | 225.011 | 1.7579 Vb | 99.62 | |
| Concatenated | Among groups | 4 | 9.713 | −0.00204 Va | −0.08 | $F_{ST}$=−0.00082 p>0.05 |
| | Within population | 116 | 287.353 | 2.47718 Vb | 100.08 | |

Significance test: 1000 permutations, df: degrees of freedom, $F_{ST}$- fixation index, Va- Variance among populations, Vb – variance within population.

cumulative variation (Fig 4A). For the *ND4* dataset, the first and second coordinates explained 11.56% and 8.14% of the total variation, respectively, with the first three axes cumulatively accounting for 27.19% (Fig 3B). In both datasets, individuals from different sampling sites showed substantial overlap, with no distinct separation among populations, consistent with weak population structuring.

## Demographic inference

Tajima's D and Fu's *Fs* showed negative values for both sequence alignments, with a significant difference for each region. All populations also exhibited significant negative Fu's and Li's D and Li's F test values (Table 1, Fig 1) and negative deviations from zero within the entire population.

The mismatch distributions of *P. argentipes* s.l. for Sri Lanka based on all five pooled study populations displayed a smooth and unimodal pattern under the sudden expansion model (Fig 4). When analyzed independently, the five *P. argentipes* s.l. populations demonstrated distinct patterns of behavior. Concatenated alignment for the Balangoda site (Fig 5) showed a unimodal distribution pattern. The other four *P. argentipes* s.l. populations exhibited a "ragged" multimodal distribution, and the results simulated a unimodal distribution of pairwise sequence differences for all lineages.

Mismatch distribution analysis (MMD) of both *COI* and *ND4* sequences revealed a multimodal pattern at the Medirigiriya and Mirigama sites (S2 Fig), suggesting complex demographic histories at these locations. The observed distributions were generally consistent with the sudden expansion model, although small but significant sum of squared deviations (SSD) were detected for *ND4* at Hambantota (SSD=0.28468, p=0.003) and Medirigiriya (SSD=0.11224, p=0.008), indicating slight deviations from the model. Overall, the MMD patterns support historical population expansion with some local complexity.

Bayesian skyline plot *COI* (Fig 6B) indicated a gradual increase in effective population size, beginning approximately 50,000 years ago, consistent with an ancient demographic expansion. In contrast, the *ND4* (Fig 6A) revealed a more pronounced increase in effective population size between 10,000 and 13,000 years ago, reflecting a more recent expansion event. The plot illustrates a significant increase in the effective population size of female *P. argentipes* s.l. in Sri Lanka, with *ND4* highlighting a more recent demographic increase compared to *COI*. BSP analysis and the resulting plot were conducted using BEAUti2, BEAST2, and Tracer v1.6. The results suggested a recent population expansion of *P. argentipes* s.l. in Sri Lanka, estimated to have occurred 10,000–13,000 years ago, coinciding with the end of the last glaciation period at the end of the Pleistocene.

## Discussion

As part of a broader investigation into the genetic landscape of *P. argentipes* s.l. in Sri Lanka, the previous study [20] focused on genetic diversity, providing foundational insights into haplotype variation and population-level diversity. The

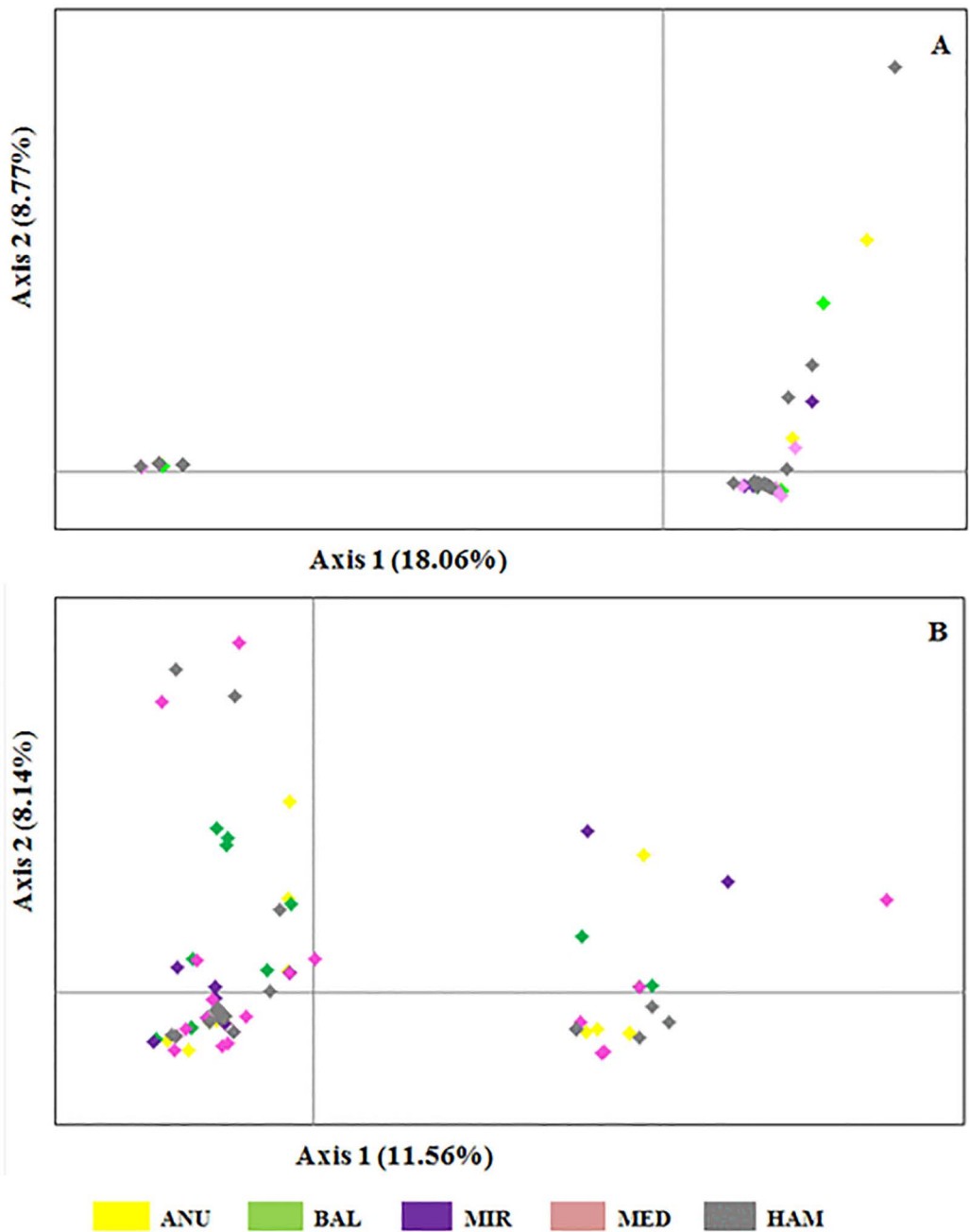

**Fig 3. Principal coordinate analysis (PCoA) scattered plot generated from genetic distance calculations using the GeneAlex package for *COI* and *ND4* partial gene sequences.** (A) *COI* data set and (B) *ND4* data set, different colored labels indicate distinct geographical origins of *P. argentipes* s.l. individuals studied.

current study expands on this work by examining population genetic structure and patterns of expansion, which were not addressed in earlier analysis. By combining these complementary perspectives, a more comprehensive understanding of the genetic dynamics of *P. argentipes* s.l. populations are presented. For clarity, key findings from the previous study are referenced where relevant to contextualize the results of this work.

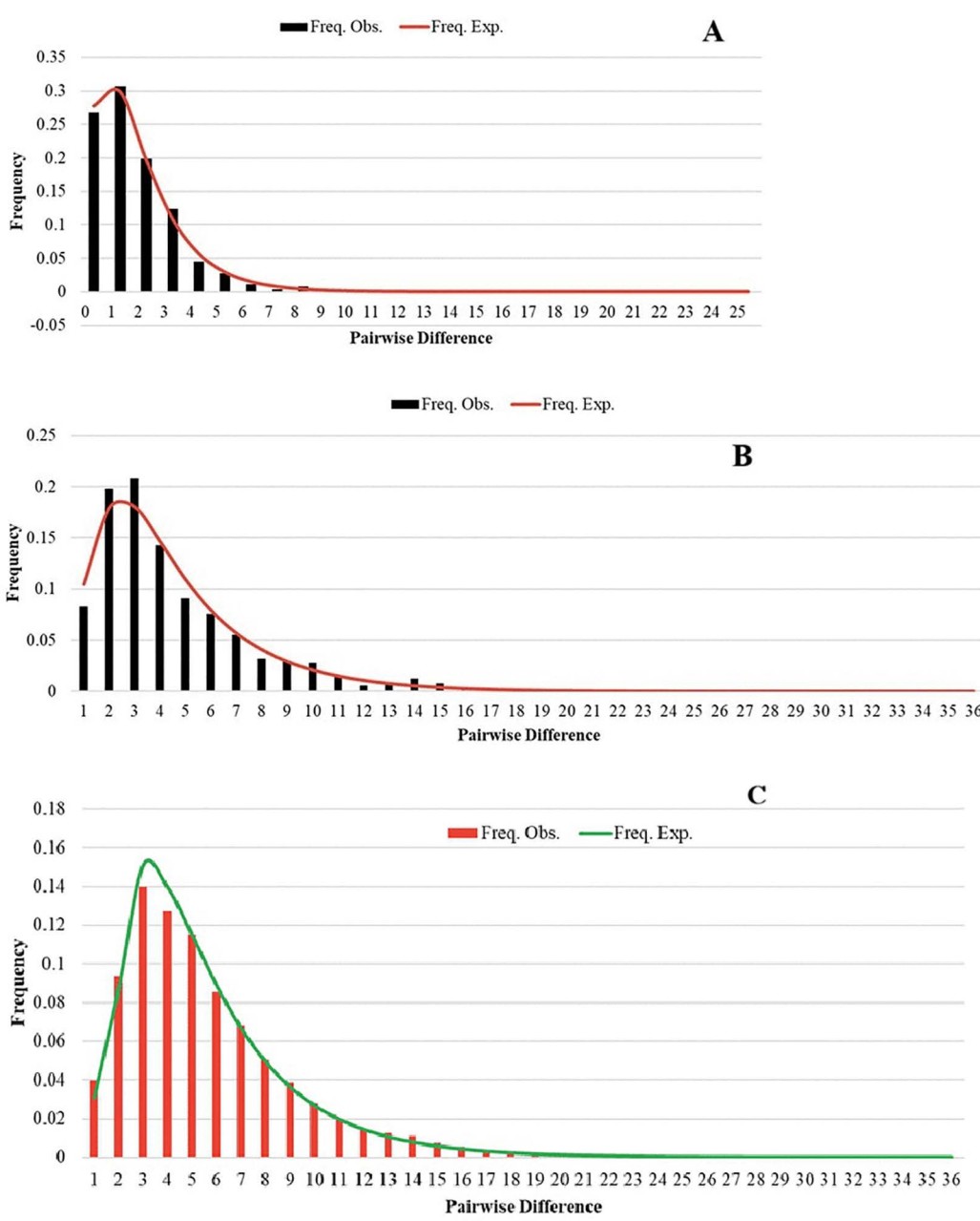

**Fig 4. Mismatch distribution analyses (MMD) of *Phlebotomus argentipes* s.l. populations, illustrating the observed and expected frequency under a sudden expansion model. (A)** Cytochrome c oxidase I (*COI*), **(B)** NADH dehydrogenase subunit 4 (*ND4*) gene sequences and (C) concatenated alignment. The observed mismatch distribution reflects historical demographic events, with deviations potentially indicating population structure or selection pressures.

*Phlebotomus argentipes* sensu lato is a species complex comprising three reproductively isolated sibling species: *Phlebotomus annandalei* (Annandale, 1910), *Phlebotomus glaucus* (Mitra and Roy, 1953), and *Phlebotomus argentipes* sensu stricto [1]. Current evidence from Sri Lanka indicates that these sibling species are not ecologically or spatially isolated and frequently co-occur at the same breeding and resting sites, exhibiting complete sympatric distribution across the country [1,2,35].

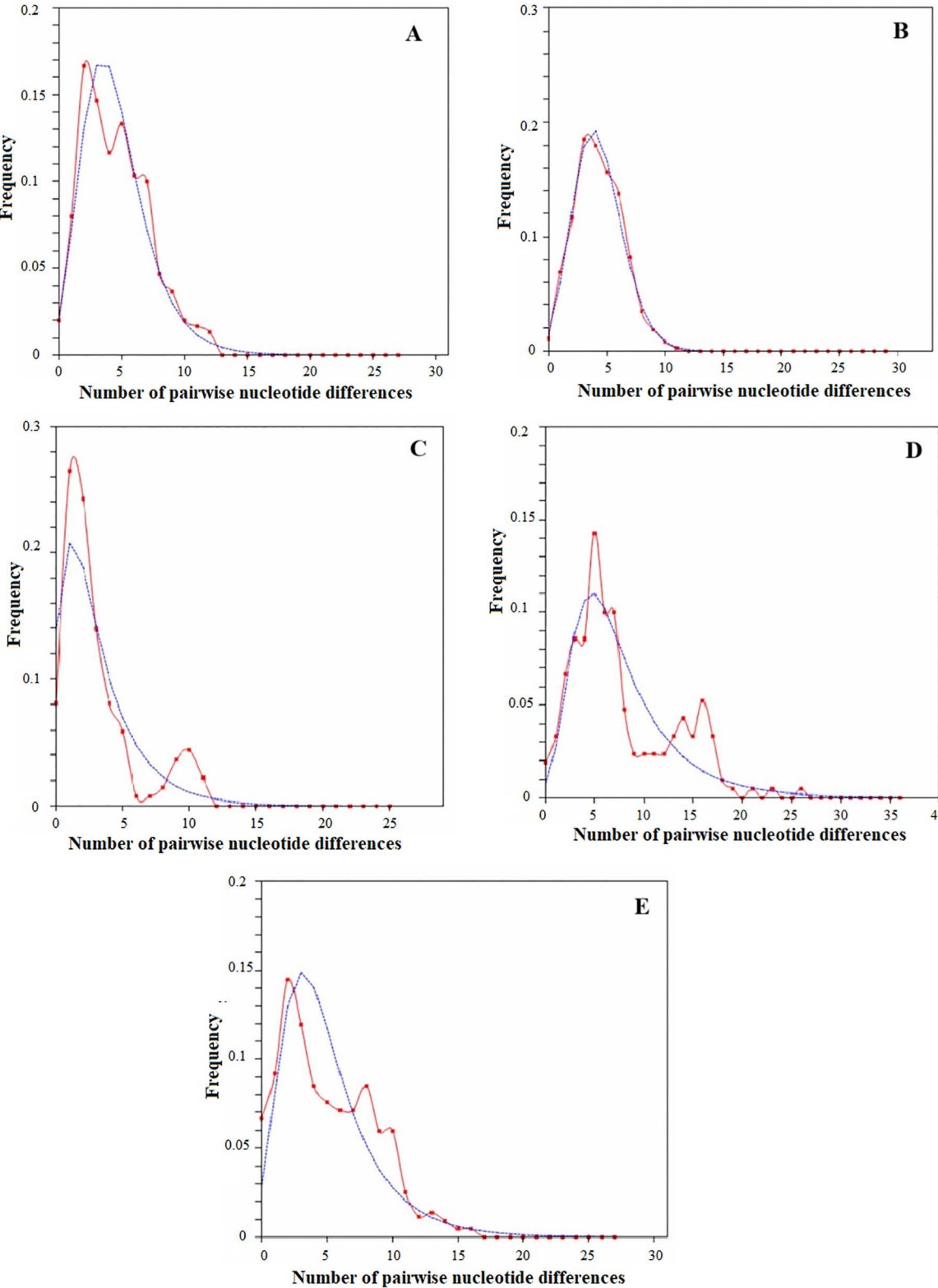

**Fig 5. Demographic expansion model comparisons based on concatenated sequence alignments of five *Phlebotomus argentipes* s.l. populations in Sri Lanka.** Observed mismatch distributions are shown in red, and expected distributions under the sudden expansion model are shown in

blue. These curves illustrate historical population dynamics, providing insights into past expansions or bottlenecks within each population. Panels: **(A)** Anuradhapura; **(B)** Balangoda; **(C)** Mirigama; **(D)** Medirigiriya; and **(E)** Hambantota.

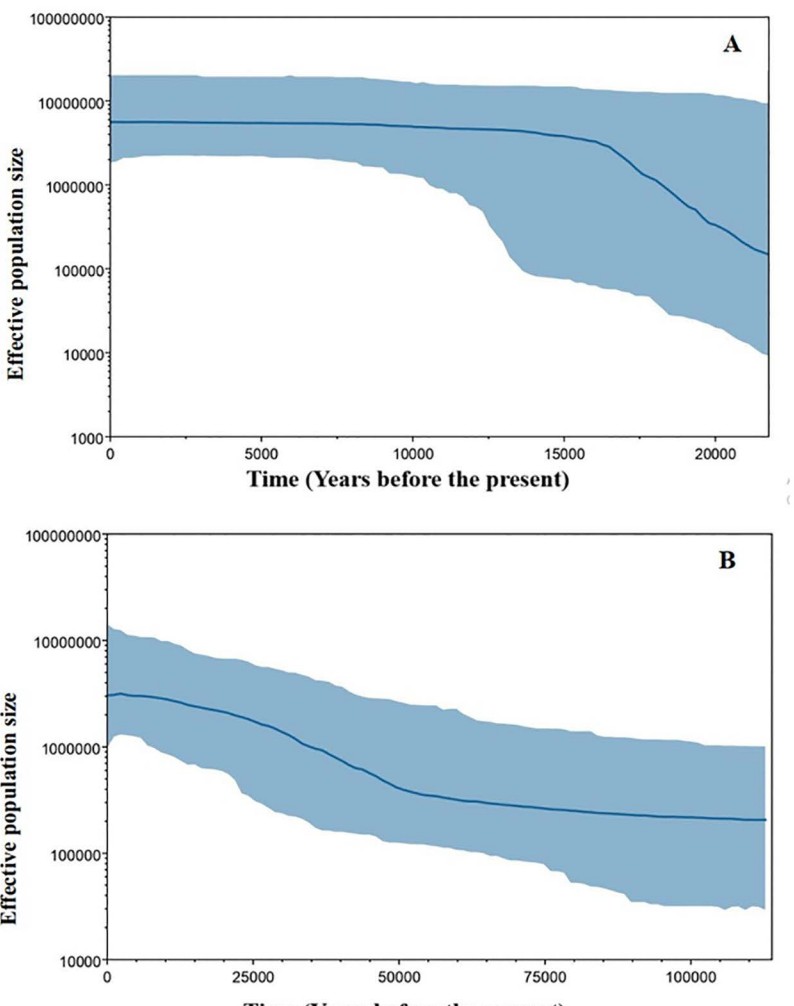

**Fig 6. Bayesian Skyline plot for the two study gene fragments.** A- *ND4,* B-*COI* and the X-axis represent the years before the present, as calculated by the Bayesian Skyline model which considers two or three sandfly generations per year, and the Y-axis indicates their effective population size. The blue area shows the minimum to maximum range for the estimated effective population.

This study acknowledges that these are distinct biological species; however, current study deliberately adopts a complex level analytical approach for several scientifically justified reasons. First, our research objectives focus on understanding the demographic history and recent population expansion patterns of the vector complex as a functional epidemiological unit. From a disease transmission perspective, all three sibling species are competent leishmaniasis vectors, and their collective demographic patterns directly influence transmission risk and vector control outcomes in Sri Lanka. Second, analyzing mitochondrial markers at the complex level is particularly appropriate for demographic inference, as mtDNA variation reflects maternal lineage history and can reveal signatures of historical population processes

(expansion, bottlenecks, gene flow) that may predate or transcend current species boundaries, especially in recently diverged taxa with incomplete lineage sorting [36]. Third, the extensive sympatric distribution of these siblings means they share environmental pressures and ecological constraints; understanding complex-wide responses to these factors provides insights into the adaptive potential and demographic resilience of the vector system as a whole [37]. Therefore, population genetic assessments should adopt a holistic approach, as analyzing sibling species separately may obscure the actual patterns of connectivity and evolutionary dynamics within the complex [38].

Independent analyses of the five *P. argentipes* s.l. populations, based on $F_{ST}$ (<0.027), indicated very low genetic differentiation between certain populations, suggesting relatively uniform dispersal among populations. Predominantly low $F_{ST}$ values suggest minimal genetic differentiation, supporting the idea of population homogeneity or recent population expansion [39], which helps in maintaining high genetic diversity across the studied sites with extensive gene flow between populations [19,20,37]. Ecological adaptability and dispersal potential may enable the vector species to sustain gene flow and haplotype diversity, reflecting the influence of strong selection pressures [40].

Differing demographic expansion timings between *ND4* (approximately 10,000–13,000 years before present) and *COI* (around 50,000 years before present) likely resulted from discrepancies between mitochondrial markers, an uncommon observation that may arise due to differences in mutation rates, selection pressures, or lineage sorting [41]. *ND4* typically evolves faster than *COI*, potentially capturing more recent demographic events, whereas *COI* may retain signatures of older expansions [20,42]. Together, these results suggest a long-term persistence of *P. argentipes* s.l. populations in Sri Lanka with evidence of both ancient and recent demographic fluctuations. The relatively short time since population expansion inferred from *ND4* may have limited the accumulation of nucleotide differences among haplotypes. In contrast, the older expansion signal observed for *COI* suggests that this marker retained deeper genealogical structure. Similar patterns of limited gene divergence following rapid expansion have also been reported in other vector species [38]. As a result, nucleotide differences between populations are minimal, and genetic variability is low, as shown in recent studies on *P. argentipes* s.l. populations in Sri Lanka [20]. These findings suggest that the recent population expansion, coupled with extensive gene flow among populations, has constrained the accumulation of nucleotide differences while maintaining moderate haplotype diversity within *P. argentipes* s.l.. Notably, the higher $F_{ST}$ values received among Balangoda and Hambantota (0.027) for *ND4*, which are CL hotspots in Sri Lanka [8], suggest a potential link between the genetic differentiation of vector populations and the spatial distribution of disease endemicity, warranting further investigation.

The phylogenetic grouping of haplotypes identified in the previous study, together with the lack of significant genealogical divergence [20], suggests that most genetic variation for both genes occurs among individuals within populations rather than between. This pattern aligns with effective local migration and/or gene flow of *P. argentipes* s.l. across Sri Lanka, potentially facilitated by the proximity of suitable habitats. Although the species' dispersal potential is constrained by its relatively short flight range (~100 m) [43] and strong wind conditions, gene flow can still occur over broader geographic scales through successive short distance movements between neighboring populations. Under typical ecological conditions, the species often remains confined to specific village microhabitats where plant and blood meal sources are locally available [44] which reduce the need for extensive dispersal and may contribute to the fine-scale genetic structures. With the scares of suitable oviposition sites and blood meals, the flight range may extend with the aided of wind currents, resulting in occasional long-distance dispersal events [15,45,46]. For example, high wind gusts- a short-lived surges in wind speed often associated with weather fluctuations [47,48], facilitate the insect dispersal in arid regions such as Medirigiriya and Hambantota [49]. While gusty winds could theoretically impose selective pressures on *P. argentipes* s.l., favoring traits that enhance flight performance, our findings suggest that gene flow remain sufficient to prevent strong genetic differentiation among populations in leishmaniasis endemic regions. This assumption aligns with the observed ragged, multimodal mismatch distribution, indicating historical bottlenecks followed by expansions. Further studies, including genomic and transcriptomic analyses, are needed to validate the influence of wind dynamics on the genetic and

functional evolution of *P. argentipes* s.l. populations. These sporadic movements, together with the absence of substantial physical barriers, can facilitate gene flow between distant populations [50,51].

The expansion of *P. argentipes* s.l. populations likely occurred alongside favorable ecological conditions in the dry zone, including increased availability of hosts and suitable microhabitats, which promoted sandfly proliferation [6,52]. Localized selective pressures such as adaptations to elevated parasitic loads and specific microhabitat conditions may drive mitochondrial genetic variation within populations, reflecting the complex interaction between demographic history and environmental factors [53].

Mantel tests for isolation-by-distance (IBD) gave mixed results, suggesting that genetic differentiation is not solely caused by geographic distance but may also involve historical factors and locus-specific dynamics [54]. The ability of *ND4* to detect spatial genetic structure demonstrates its effectiveness in capturing current gene flow dynamics [23]. These differences between markers probably result from their distinct evolutionary rates and sensitivities to genetic and demographic processes [55], emphasizing the importance of using multiple genetic markers.

Mismatch distribution analysis for *P. argentipes* s.l. in Sri Lanka showed a unimodal pattern for the overall population, indicating recent demographic expansion [30]. In contrast, *P. argentipes* s.l. populations exhibited multimodal distributions, suggesting demographic stability or multiple localized expansion events [56]. These patterns may be influenced by factors such as seasonal environmental changes, host availability, and parasite prevalence, all of which affect sandfly reproduction and dispersal. Tajima's D values suggest that, despite some genetic structuring – possibly due to historical divergence or ecological factors- recent population expansion and gene flow have facilitated admixture and genetic connectivity among populations, blurring strict geographic genetic boundaries [54,57]. Understanding the dynamics of these processes is important for gaining a comprehensive perspective on the connectivity and genetic structure of *P. argentipes* s.l. populations, which is essential for designing targeted vector control strategies. Although insecticide-driven selection may contribute to the population structure and expansion of *P. argentipes* s.l., it is unlikely to act alone. The influence of human mobility-particularly through migrant labor between India and Sri Lanka-as well as the role of mobile animal reservoirs such as dogs should also be considered. These factors may facilitate the introduction or spread of both the vector and the parasite, contributing to the emergence of leishmaniasis in new regions.

Current study focused on *P. argentipes* sensu lato as a species complex, within which morphologically similar sibling species coexist sympatrically at most collection sites. Given this overlapping distribution and the frequent occurrence of mixed populations, treating the complex as a single analytical unit provides a realistic representation of natural gene flow and population connectivity. Nevertheless, as sibling species may differ in vectorial competence and ecological traits, future studies incorporating molecular species-level identification are warranted to disentangle intra-complex variation and refine the understanding of leishmaniasis transmission dynamics.

## Conclusions

The lack of distinct genetic structure within *P. argentipes* s.l., indicated by the absence of genetic clustering or drift, suggests high connectivity among populations in Sri Lanka. The moderate to high genetic diversity observed may support the survival of vector populations across different regions. The patterns of population expansion, together with genetic homogeneity, could create conditions conducive to the ongoing transmission of leishmaniasis, especially in areas with high vector prevalence. Furthermore, this study demonstrates the effectiveness of the *ND4* marker and concatenated analysis in assessing population dynamics and genetic trends in *P. argentipes* s.l., highlighting its usefulness for future studies on population genetics and vector surveillance.

## Supporting information

**S1 Table. *Phlebotomus argentipes* s.l. sandfly collection record from five different collection sites in Sri Lanka.**
(DOCX)

**S1 Fig. Inter-haplotype pairwise distance matrices of sandfly populations from five sampling sites based on concatenated *COI* and *ND4* sequences.**
(TIF)

**S2 Fig. Comparison of demographic inference expansion models based on *ND4* and *COI* sequences across five *Phlebotomus argentipes* s.l. populations in Sri Lanka.** The X-axis represents the number of pairwise nucleotide differences, and the Y-axis indicates their frequency. Panels (a, b) correspond to Anuradhapura; (c, d) Balangoda; (e, f) Mirigama; (g, h) Medirigiriya; and (i, j) Hambantota. *COI*-based observed and expected demographic expansions are represented in blue and red, respectively, while *ND4*-based observed and expected expansions are shown in green and yellow, respectively. These graphs illustrate historical population dynamics, providing insights into past expansions and bottlenecks within each population.
(TIF)

## Acknowledgments

The Epidemiology Unit, Ministry of Health, Sri Lanka, the Anti-Malaria Campaign, and the regional public health officers of the sampling areas are acknowledged for their generous provision of leishmaniasis disease prevalence data and their invaluable assistance during the fieldwork.

## Author contributions

**Conceptualization:** B. G. D. N. K. De Silva, Iresha N. Harischandra.

**Data curation:** W. M. M Wedage.

**Formal analysis:** W. M. M Wedage, Iresha N. Harischandra.

**Funding acquisition:** B. G. D. N. K. De Silva.

**Investigation:** W. M. M Wedage, Iresha N. Harischandra, B. G. D. N. K. De Silva.

**Methodology:** W. M. M Wedage, Iresha N. Harischandra, B. G. D. N. K. De Silva.

**Project administration:** B. G. D. N. K. De Silva.

**Resources:** B. G. D. N. K. De Silva.

**Software:** W. M. M Wedage.

**Supervision:** B. G. D. N. K. De Silva, Iresha N. Harischandra, O. V. D. S. J. Weerasena, S. A. S. C. Senanayake.

**Writing – original draft:** W. M. M Wedage.

**Writing – review & editing:** B. G. D. N. K. De Silva, I. N. Harischandra, O. V. D. S. J. Weerasena, S. A. S. C. Senanayake.

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
