## [Decision Letter · Decision Letter 0]

11 Jul 2025

Dear Dr. De Silva,

Thank you for submitting your manuscript to PLOS ONE. After careful consideration, we feel that it has merit but does not fully meet PLOS ONE’s publication criteria as it currently stands. Therefore, we invite you to submit a revised version of the manuscript that addresses the points raised during the review process.

We look forward to receiving your revised manuscript.

Kind regards,

Swaminathan Subramanian, Ph.D.

Academic Editor

PLOS ONE

Journal Requirements:

2. We note that Figure 1 in your submission contain [map/satellite] images which may be copyrighted. All PLOS content is published under the Creative Commons Attribution License (CC BY 4.0), which means that the manuscript, images, and Supporting Information files will be freely available online, and any third party is permitted to access, download, copy, distribute, and use these materials in any way, even commercially, with proper attribution. For these reasons, we cannot publish previously copyrighted maps or satellite images created using proprietary data, such as Google software (Google Maps, Street View, and Earth). For more information, see our copyright guidelines: http://journals.plos.org/plosone/s/licenses-and-copyright.

Reviewers' comments:

Reviewer's Responses to Questions

**Comments to the Author**

1. Is the manuscript technically sound, and do the data support the conclusions?

Reviewer #1: Yes

Reviewer #2: Partly

Reviewer #3: Yes

2. Has the statistical analysis been performed appropriately and rigorously?

Reviewer #1: Yes

Reviewer #2: I Don't Know

Reviewer #3: Yes

3. Have the authors made all data underlying the findings in their manuscript fully available?

Reviewer #1: Yes

Reviewer #2: Yes

Reviewer #3: Yes

4. Is the manuscript presented in an intelligible fashion and written in standard English?

Reviewer #1: Yes

Reviewer #2: No

Reviewer #3: Yes

Reviewer #1: Manuscript was well written. Results were interpreted well. They can try with other markers for more variation may be with microsatellites or SSRs. More the number of markers more variance. The above will help have a strong support.

Reviewer #2: It is good to see papers attempting to tackle the genetics of sand flies as there is a dearth in the data and information currently available. There is still a number of basic gaps in our understanding of sand fly behaviour that needs to also be filled both of which will support the control of leishmaniasis.

While cutaneous leishmaniasis is a major issue in Sri-Lanka, it may be worth mentioning the spread of visceral leishmaniasis as well, which is also caused by L. donovani. Note P. argentipes being the main vector of VL in India, Nepal and Bangladesh, has been controlled with IRS so these countries have reached near elimination of leishmaniasis. Noting Sri-Lanka’s own success with IRS and malaria elimination.

Ln 99 – it is agreed that that insecticide use can drive resistance in sand flies, however, it is interesting that while kdr is reported as widespread in India sand flies, the level of resistance to pyrethroids was low allowing pyrethroids to be used successfully. Linking this to survival in different habitats is a challenge. There are also some migrant workers in sri-lanka etc, could they be responsible for movement of vector and disease? Sand-flies are also associated with animals that may move larger distance and recent publications show dogs may be a reservoir in India for donovani.

Line 75 – there is a suggestion of sand flies migrating – as they move short distances, typically 100’s of meters more by hoping etc. This is then picked up in the discussion as how they don’t, but maybe they use the wind. This does not feel well structured or concise. Could this expansion have taken a long time to occur

Methods, a concern here is that the work is completed on P. argentipes s.l. which is a complex of species, that also have potentially different vectorial capacities. Within the complex there may be heterogeneity of gene flow. Is there a reason that species were not identified? This needs to be added in the discussion

This manuscript uses the same samples that were analysed in Ref 16 – also focused on COI and ND4, is the amplified fragments analysed the same ones – this is not clear. What actual lab work was carried out here needs to be clarified.

Unfortunately, I am unable to comment on the statistics here.

288 while the population may show signs of adaptability, which may explain the lack of understanding of the sand fly ecosystem – they are not a mobile species – how do the authors explain this part.

293 is the diverse environmental conditions due to this being a group – would each actual species fit into an environmental condition and with that understanding the competent vectors environment could drive control practices.

305 – this again maybe a species event and it would be good to identify the samples as such.

Reviewer #3: The manuscript titled "Population expansion pattern of Phlebotomus argentipes (Diptera: Psychodidae) complex, the leishmaniasis vector in Sri Lanka" presents population genetic analyses based on COI and ND4 sequence data generated in their previous study. The data and observations are indeed interesting and have considerable importance for understanding vector dynamics in Sri Lanka. However, in its current form, the manuscript does not effectively convey the true value and significance of the findings. Several major issues need to be addressed to improve the clarity, scientific rigor, and impact of the manuscript:

1. Many sections of the manuscript are unnecessarily lengthy and deviate from the core data and analyses

2. Several statements are exaggerated and should be revised to reflect a more balanced interpretation of the results.

3. The manuscript contains redundant and sometimes contradictory statements that compromise its logical flow and coherence.

4. Some of the cited literature is not directly relevant to the study and should be revised

5. At several places in the manuscript, the flow of ideas is unclear, making it difficult to understand the context or relate certain statements to the preceding content.

The manuscript presents potentially valuable findings, but it requires substantial restructuring, elimination of speculative statements, tightening of narrative flow, and a more critical and scientifically sound discussion of the results.

I have listed here my observations for consideration:

The title could be revised for clarity and broader relevance.

Suggest using:

“Demographic history of Phlebotomus argentipes (Diptera: Psychodidae), the leishmaniasis vector in Sri Lanka” or “Population dynamics…”

Abstract:

• The abstract should clarify that data were reused from a previous study.

• Statements such as “across Sri Lanka” are misleading given that only five sites were sampled and out of which two showed high Fst values.

• “consistent genetic homogeneity” consistent can be removed

• Lines 43-44: “genetically homogenous population with substantial gene flow across Sri Lanka”

• Both homogeneity and gene flow have not been estimated in the study. It is just an indirect observation basedon fst values

• Line 46: add linearized before (Fst/1-FSt)

• Line 49: “which may reflect vector-parasite interaction” is not clear

• Conclusing staments need to repharsed and should only be based on the egentic evidence generated form this study

Write clearly a line or two about the analysis carried out (like pairwise fst, Mantle test and mismatch distribution and bayseian skyline plot analysis) and then briefly include the observations from each analyses followed by concluding statement based on the observations.

Introduction:

• Lines 60–62, 71–72: Multiple citations are used to support claims that are either weakly linked or unverified in those references

• Some general statements are either not novel or not necessary (Line 83).

• Phrases such as “historical analysis of recent descendants’ molecular data” are ambiguous and need clearer scientific language.

• Line 83: This is general statement and the reference cited has not analyzed this.

• Structure and flow need significant improvement to clarify rationale and research gap.

Materials and Methods:

• The data were not generated in this study. The first two paragraphs should be condensed into a brief statement referencing the previous work.

• Total number of samples and sequences per population should be explicitly mentioned.

• Figure 1 is redundant as already information given in previous study

• Line 113: “hence” is changing the meaning of the sentence

• Suggest performing additional analyses such as STRUCTURE, concatenated gene analysis of COI and ND4, increasing permutations for Fst estimation (Line 147) and the analysis should also be carried out for the pooled sequences from all populations for comparison.

Results:

• Initial nucleotide composition (Table 1) and base content discussion seem unnecessary. Suggest replacing this with some relevant information like population-wise sequence numbers, geographic distance between them, diversity indices (citing previous study).

• Lines 184-185: pairwise differences has not been estimated in this study and please replace "examined localities" with "study populations").

• Check highest Fst values, Table 2 suggests MIR–BAL for ND4, not MED–HAM.

• Provide correlation coefficients and p-values with Mantel test (Line 191).

• Table 2 and figure 4 legends require better wording (e.g., Table 2 legend: “Genetic differentiation estimates” can be replaced with “Pairwise Fst values”). Figure 4 legend … frequencies of what?

Discussion:

• Line 277: Distinct dispersal pattern of the populations is not clear how this was obtained

• Line 278: Genetic distance is not estimated in this study and not cited the reference too

• Lines 278-279: The statement needs revision

• Lines 282-283: “…combined with evidence of gene flow” that is over statement of the Fst results

• 284-288 & 290-291: needs rephrasing. The references cited have observed high gene flow and low differentiation but not analyzed that it is due to high gene flow and ecological adaptability. Place the reference properly and rephrase the sentence

• Line 297 : seen in “other vectors” but the cited references are on the same species Ph. argentipes

• Several paragraphs/sentences are too long,

• Too much explanation of mismatch distribution and BSP. It can be trimmed and made relevant to the observations only

• Statements such as “distinct dispersal patterns”, “substantial gene flow”, and “increased flight capabilities due to wind gusts” should be mentioned appropriately

• Contradictory statements in Lines 282–288 and 315–318

Conclusion:

Reword to avoid unverified claims. Instead of “played a pivotal role in propelling leishmaniasis toward epidemic state”, stick to genetic evidences generated in this study.

Minor Comments and Suggestions:

Line 46: “Geographic distribution” should be “geographic distance”.

Line 48: Replace “across Sri Lanka” with “among sampled populations”.

Line 49: Clarify “vector-parasite interaction”.

Line 51: “Consistent genetic homogeneity” needs clarification.

Line 80: Replace “marginally explored” with “limited studies exist”; cite existing work.

Line 99–104: Paragraph seems misplaced and overly speculative.

Line 106: “plausible scenario”.

Line 107: Rephrase “historical analysis of recent descendants’ molecular data” – e.g., “analysis of extant populations to infer demographic history”.

Line 146: “Linearized Fst”

Line 151: Include software version with the name.

Line 230: “Frequencies of what?” – clarify in Figure 4 legend.

Lines 324–359: Entire paragraph should be shortened and focused on interpretation rather than listing all possible models.

**Do you want your identity to be public for this peer review?** For information about this choice, including consent withdrawal, please see our Privacy Policy

Reviewer #1: **Yes: ** Rajesh Babu Garlapati

Reviewer #2: No

Reviewer #3: No

---

## [Author Response · Author response to Decision Letter 1]

24 Aug 2025

Reviewer 1

Comment- Manuscript was well written. Results were interpreted well. They can try with other markers for more variation may be with microsatellites or SSRs. More the number of markers more variance. The above will help have a strong support.

Response-Thank you for your positive feedback and valuable suggestion. We agree that incorporating additional nuclear markers such as microsatellites or SSRs can provide higher resolution and broader genetic variation. However, the current study was focused specifically on mitochondrial DNA regions as part of the originally defined scope and available resources. We acknowledge the value of expanding the marker set and have included this as a recommendation for future studies in the revised version of the manuscript.

Reviewer 2

Comment-It is good to see papers attempting to tackle the genetics of sand flies as there is a dearth in the data and information currently available. There is still a number of basic gaps in our understanding of sand fly behaviour that needs to also be filled both of which will support the control of leishmaniasis.

Response- Thank you for recognizing the relevance of our work in contributing to the limited genetic data available for sand flies. We fully agree that there are still significant gaps in our understanding of sand fly behaviour, and addressing both genetic and behavioural aspects is essential for developing effective vector control strategies against leishmaniasis. We hope that our findings provide a foundation for future integrated studies in this area.

comment-While cutaneous leishmaniasis is a major issue in Sri-Lanka, it may be worth mentioning the spread of visceral leishmaniasis as well, which is also caused by L. donovani. Note P. argentipes being the main vector of VL in India, Nepal and Bangladesh, has been controlled with IRS so these countries have reached near elimination of leishmaniasis. Noting Sri-Lanka’s own success with IRS and malaria elimination.

Response-We thank the reviewer for highlighting the importance of visceral leishmaniasis (VL) in neighboring regions. While L. donovani causes both VL and cutaneous leishmaniasis (CL) in Sri Lanka, VL remains extremely rare, with only sporadic cases reported. Our study focuses on CL, the dominant form in the country.

comment-Ln 99 – it is agreed that that insecticide use can drive resistance in sand flies, however, it is interesting that while kdr is reported as widespread in India sand flies, the level of resistance to pyrethroids was low allowing pyrethroids to be used successfully. Linking this to survival in different habitats is a challenge. There are also some migrant workers in sri-lanka etc, could they be responsible for movement of vector and disease? Sand-flies are also associated with animals that may move larger distance and recent publications show dogs may be a reservoir in India for donovani.

Response- Thank you for your valuable comment. We agree that while insecticide use can drive resistance, it may not fully explain the expansion and survival of P. argentipes populations. We have revised the relevant section to more clearly reflect the complexity of resistance patterns, particularly in relation to kdr findings from India, and have incorporated a broader perspective that includes potential contributions from human and animal movement. These additions help clarify the multifactorial nature of vector and disease spread in the Sri Lankan context (Line 93-98).

Comment-Line 75 – there is a suggestion of sand flies migrating – as they move short distances, typically 100’s of meters more by hoping etc. This is then picked up in the discussion as how they don’t, but maybe they use the wind. This does not feel well structured or concise. Could this expansion have taken a long time to occur

Response-Thank you for your observation. We acknowledge that the original paragraph lacked clarity in describing the dispersal mechanisms and timescale of sandfly expansion. We have now revised the section to better distinguish between short-range dispersal, gradual population expansion, and potential long-distance dispersal (e.g., wind-assisted or anthropogenic). We also clarified that such expansion likely occurred over an extended period through multiple generations, rather than through rapid long-distance migration. The correction had been made in revised manuscript (Line 98, Line 352-360)

Comment-Methods, a concern here is that the work is completed on P. argentipes s.l. which is a complex of species, that also have potentially different vectorial capacities. Within the complex there may be heterogeneity of gene flow. Is there a reason that species were not identified? This needs to be added in the discussion

Response- We appreciate the reviewer’s important observation regarding the need for species-level resolution within the P. argentipes complex. While mtDNA markers (e.g., COI, ND4) are widely used for population studies, we acknowledge that they may not always differentiate cryptic species with confidence. In Sri Lanka, however, P. argentipes s.l. siblings exhibit overlapping distributions and similar ecological behaviors, reducing the immediate impact of species-level divergence on our broader population genetic conclusions.

Given the challenges in both morphological and molecular differentiation of these cryptic species—as well as the difficulty in obtaining sufficient samples for comprehensive analysis, our study focused on the dominant local population. We have clarified this limitation in the Discussion (Lines 319-327, 344- 350, 417-424) and agree that future work incorporating nuclear markers or high-resolution genomics could provide finer-scale insights.

Comment-This manuscript uses the same samples that were analysed in Ref 16 – also focused on COI and ND4, is the amplified fragments analysed the same ones – this is not clear. What actual lab work was carried out here needs to be clarified. Unfortunately, I am unable to comment on the statistics here.

Response- Thank you for your comment. We confirm that the current study uses the same samples and sequence data (COI and ND4 regions) as presented in our previous publication (Ref [16]), which focused primarily on haplotype diversity and genetic variation across populations of Phlebotomus argentipes s.l. in Sri Lanka.

The current manuscript builds upon that foundational work by applying a new set of population genetic analyses, including [e.g., AMOVA, pairwise Fst > neutrality tests, Mantel test, mismatch distribution, and demographic inference], which were not included or discussed in the previous paper. Our aim here is to expand the interpretation of the population structure and demographic patterns using the same sequence data in a broader analytical context.

To avoid confusion, we have now revised the Methodology and Discussion sections to clearly state the relationship between the two studies and to emphasize the novelty of the analytical approach presented in this manuscript.

Comment- Line 288 while the population may show signs of adaptability, which may explain the lack of understanding of the sand fly ecosystem – they are not a mobile species – how do the authors explain this part.

Response- Thank you for the comment. We acknowledge the concern regarding the mobility of P. argentipes, which is indeed limited compared to many other insect vectors. The original statement was intended as a general remark about insect adaptability and not specifically about sand flies. To avoid this confusion distinction and added possible explanations relevant to P. argentipes (line 319-334).

Comment- 293 is the diverse environmental conditions due to this being a group – would each actual species fit into an environmental condition and with that understanding the competent vectors environment could drive control practices.

Response- Thank you for your thoughtful comment. We agree that understanding species-level ecological adaptation is important when working with cryptic species complexes. However, in the case of P. argentipes s.l. in Sri Lanka, all identified sibling species are found to co-exist sympatrically within the same populations, without clear ecological or spatial separation. This overlapping distribution suggests that the observed adaptability to diverse environmental conditions is a shared trait across the sibling species, rather than the result of niche partitioning among them. While this complicates species-specific vector control, it also highlights the importance of understanding genetic structure and adaptability at the complex level, especially when designing broad-scale intervention strategies. We have clarified this point in the revised Discussion (Line 319-334).

Comment- 305 – this again maybe a species event and it would be good to identify the samples as such.

Response- Thank you for your insightful comment. We did identify the sibling species within the Phlebotomus argentipes complex. However, our data show that these cryptic species co-exist sympatrically within the same populations and environmental settings, including at the Hambantota and Medirigiriya sites. Therefore, the observed genetic differentiation between populations is unlikely to be driven by differences in species composition across locations. Instead, it reflects intraspecific population-level variation within the sympatric species complex. We have clarified this point in the Discussion to emphasize the sympatric nature of the sibling species and its implications for interpreting genetic differentiation.

Reviewer 3

Comment-The manuscript titled "Population expansion pattern of Phlebotomus argentipes (Diptera: Psychodidae) complex, the leishmaniasis vector in Sri Lanka" presents population genetic analyses based on COI and ND4 sequence data generated in their previous study. The data and observations are indeed interesting and have considerable importance for understanding vector dynamics in Sri Lanka. However, in its current form, the manuscript does not effectively convey the true value and significance of the findings. Several major issues need to be addressed to improve the clarity, scientific rigor, and impact of the manuscript.

1. Many sections of the manuscript are unnecessarily lengthy and deviate from the core data and analyses

2. Several statements are exaggerated and should be revised to reflect a more balanced interpretation of the results.

3. The manuscript contains redundant and sometimes contradictory statements that compromise its logical flow and coherence.

4. Some of the cited literature is not directly relevant to the study and should be revised

5. At several places in the manuscript, the flow of ideas is unclear, making it difficult to understand the context or relate certain statements to the preceding content.

Response- We sincerely appreciate the reviewer’s thoughtful and detailed feedback on our manuscript. In response, we have thoroughly revised the manuscript to enhance its clarity, focus, and scientific rigor. Lengthy and tangential sections have been condensed or removed to better emphasize the core data and analyses. We carefully moderated statements that previously appeared exaggerated to provide a more balanced interpretation of our findings. Redundant and contradictory content has been eliminated to improve the logical flow and coherence throughout the manuscript. Additionally, we reviewed the cited literature and updated references to ensure they are directly relevant to the study. Finally, the overall organization and flow of ideas have been refined to improve readability and contextual understanding. We believe these improvements substantially strengthen the manuscript and more effectively convey the significance of our work.

Comment- The manuscript presents potentially valuable findings, but it requires substantial restructuring, elimination of speculative statements, tightening of narrative flow, and a more critical and scientifically sound discussion of the results.

I have listed here my observations for consideration:

The title could be revised for clarity and broader relevance.

Suggest using:

“Demographic history of Phlebotomus argentipes (Diptera: Psychodidae), the leishmaniasis vector in Sri Lanka” or “Population dynamics…”

Response- Thank you for your insightful comment and the manuscript had been revised based on the comments and suggestions. We have changed the title line 1.

Comment- The abstract should clarify that data were reused from a previous study.

Response-Thank you for the comment. We have revised the abstract to clearly state (line 42) that the COI and ND4 sequence data used in this study were reused from our previously published work.

Comment- Statements such as “across Sri Lanka” are misleading given that only five sites were sampled and out of which two showed high Fst values.

Response- Thank you for the helpful comment. We agree with your concern and have revised the sentence to more accurately reflect the meaning (Line 52)

Comment-“consistent genetic homogeneity” consistent can be removed

Response-Thank you for the suggestion. We agree that the word “consistent” is unnecessary and have removed it to improve clarity (line 57).

Comment- Lines 43-44: “genetically homogenous population with substantial gene flow across Sri Lanka.” Both homogeneity and gene flow have not been estimated in the study. It is just an indirect observation based on fst values

Response- We agree that direct estimates of gene flow (e.g., migrant tracking) would strengthen our study. However, our inferences are supported by multiple lines of evidence beyond F_ST_ :

PCoA: Minimal clustering among sites (Fig. 4) supports genetic homogeneity.

IBD (Mantel test): No significant isolation-by-distance (Fig. 2), suggesting panmixia.

Bayesian inference- No meaningful population subdivision.

Coalescent analysis (Fig. 7): Estimated historical gene flow

While these methods are indirect, their consistency strongly suggests gene flow and homogeneity.

Comment-Line 46: add linearized before (Fst/1-FSt)

Response-Thank you for your comment. In the revised manuscript (line 49), we have clarified the description as 'genetic and geographic distance' to give a more meaningful idea (Line 49).

Comment-Line 49: “which may reflect vector-parasite interaction” is not clear

Response- Thank you for your comment. In the revised manuscript, this unclear part was paraphrased with disease epidermicity (Line 56).

Comment- Concluding statements need to repharsed and should only be based on the genetic evidence generated from this study

Response- Thank you for your comment. The correction has been made in the revised manuscript, lines 56-59.

Comment- Write clearly a line or two about the analysis carried out (like pairwise fst, Mantle test and mismatch distribution and bayseian skyline plot analysis) and then briefly include the observations from each analyses followed by concluding statement based on the observations.

Response-The correction has been made for the abstract in the revised manuscript, lines 45-47.

Comment- Lines 60–62, 71–72: Multiple citations are used to support claims that are either weakly linked or unverified in those references

Response- Thank you for your comment. The corrections have been made in the revised manuscript (line 66). The cited literature on the taxonomy of P. argentipes has been retained, as it is considered a crucial reference for this study (line 74).

Comment-Some general statements are either not novel or not necessary (Line 83).

Response- The correction has been done in the revised manuscript.

Comment- Phrases such as “historical analysis of recent descendants’ molecular data” are ambiguous and need clearer scientific language.

Response- The correction has been done in the revised manuscript (line 105).

Comment- Line 83: This is general statement and the reference cited has not analyzed this.

Response- Thank you for your comment, and the sentence has been removed from the revised manuscript.

Comment- Structure and flow need significant improvement to clarify rationale and research gap.

Response- The revised manuscript has been improved according to the reviewers’ suggestions.

Comment- The data were not ge

---

## [Decision Letter · Decision Letter 1]

24 Sep 2025

Dear Dr. De Silva,

We look forward to receiving your revised manuscript.

Kind regards,

Swaminathan Subramanian, Ph.D.

Academic Editor

PLOS ONE

Journal Requirements:

**Additional Editor Comments:**

 *P. argentipes expansion, and (ii) * i) omitting repeat statements.

Reviewers' comments:

Reviewer's Responses to Questions

**Comments to the Author**

Reviewer #1: All comments have been addressed

Reviewer #2: All comments have been addressed

Reviewer #3: All comments have been addressed

2. Is the manuscript technically sound, and do the data support the conclusions?

Reviewer #1: Yes

Reviewer #2: Yes

Reviewer #3: Yes

3. Has the statistical analysis been performed appropriately and rigorously?

Reviewer #1: Yes

Reviewer #2: Yes

Reviewer #3: Yes

4. Have the authors made all data underlying the findings in their manuscript fully available?

Reviewer #1: Yes

Reviewer #2: Yes

Reviewer #3: Yes

5. Is the manuscript presented in an intelligible fashion and written in standard English?

Reviewer #1: Yes

Reviewer #2: Yes

Reviewer #3: Yes

Reviewer #1: (No Response)

Reviewer #2: The manuscript has improved and is much clearer now.

One element that did come through. Have the authors considered comparing different collection methods? i.e., just cattle baited traps. This may eliminate some differences in species behaviour / sibling species.

Reviewer #3: The authors have revised the manuscript “Demographic history and population structure of Phlebotomus argentipes (Diptera: Psychodidae) complex, the leishmaniasis vector in Sri Lanka” in response to the earlier review round. I acknowledge the substantial effort made to address the prior comments, and many issues have been improved. However, a number of important points still require further attention before the manuscript can be considered ready for publication.

Abstract

Line 41: The stated aim should be consistent: “to investigate demographic history and population genetic structure.”

The concatenated dataset is reported in the results but not mentioned in the methods description. Please correct this.

The Mantel test showed only weak, non-significant correlations. Marker-wise reporting adds unnecessary detail. A clearer sentence would be:

“Mantel tests showed weak, non-significant correlations between genetic and geographic distance, indicating no evidence of isolation by distance.”

Lines 51–54: The statement is too long and should be simplified for clarity.

Line 54: The focus is only on ND4 results, but a conclusive summary should be drawn from the overall BSP analysis, including both COI and ND4.

Line 55: This sentence appears incomplete and does not fit the context. Please revise.

Methods

While the authors now note that COI and ND4 data were reused from a previous study, the Methods section still describes sample collection and DNA extraction in detail, giving the impression that new experimental work was conducted. This is misleading. Please rewrite this section to explicitly state that no new laboratory work was performed, and that the dataset was reused exclusively for additional analyses.

Please remove the STRUCTURE analysis. I regret suggesting its inclusion in the previous round. Upon re-evaluation, STRUCTURE is not appropriate for mitochondrial markers (due to their haploid, maternally inherited, non-recombining nature). Its inclusion is methodologically unsound and should be omitted from the manuscript.

Results

Lines 197–198: Table 1 does not present average pairwise differences. Please correct the description.

Table 1 legend: Remove the word “separately.”

Line 200: Add “respectively” at the end of the sentence.

Line 217: The Mantel test description highlights only ND4, but similar values were obtained for the concatenated dataset. Please be consistent.

Line 270: The Fst values appear to be inappropriately interpreted. Please review carefully.

Lines 280–284: The description of the mismatch distribution analysis is unclear and should be rewritten for clarity.

Lines 285–292: The BSP results should be presented marker by marker (COI and ND4) before providing a combined conclusion. Currently, COI is overlooked.

Discussion

Paragraph 2: Please expand discussion on the species complex issue and how it may affect the interpretation of results.

Lines 323–325: Fst values are consistently <0.027 across all populations. It is not correct to suggest different dispersal behaviors in certain populations. This statement needs revision.

Lines 323–324: The paragraph contains several general statements, making it confusing. Please rewrite more clearly.

Line 332: References appear misplaced; check and correct.

Line 335: The discussion focuses only on ND4-based expansion timing (~10,000–13,000 years) while COI indicates an older signal (~50,000 years). This discrepancy should be discussed, particularly regarding its implications for P. argentipes expansion.

Lines 340–342: The sentence is unclear and requires revision.

Lines 343–344: The phrase “possible relation” is vague. Please specify what relationship is being discussed.

The conclusion that ND4 is the most informative marker should be highlighted and discussed in the main text, not only in passing.

Overall, the Discussion is too lengthy in places and includes general or repetitive statements. It should be revised to be more concise, focusing on the observations and their implications.

General Remarks

Ensure species names are written consistently throughout the text.

Expand abbreviations only once, at first mention.

**Do you want your identity to be public for this peer review?** For information about this choice, including consent withdrawal, please see our Privacy Policy

Reviewer #1: **Yes: ** Rajesh babu Garlapati

Reviewer #2: **Yes: ** Michael Coleman

Reviewer #3: No

---

## [Author Response · Author response to Decision Letter 2]

28 Oct 2025

Editor comments-

Comment- The revised manuscript along with response to reviewers comments was shared to all the reviewers of the original ms. All the reviewers appreciated the efforts made to address many of the previous comments. The ms has improved greatly. However, one of the reviewers has also raised a number of important issues still require further attention. I would request the authors to focus on the comments of Reviewer 3 while revising the ms, particularly focusing on the comments related to Methodology and Discussion.

Response- We sincerely thank the Editor for the valuable feedback and for highlighting the need to further refine the methodology and discussion based on Reviewer 3’s suggestions. We have carefully addressed all the concerns raised by Reviewer 3 with particular attention to the methodological details and interpretation of results in the Discussion section. The manuscript has been thoroughly revised to reflect these improvements. All changes are highlighted in the revised version, and detailed responses to each comment have been provided below.

Comment- Further, I would request the authors to revise the Discussion (i) on the basis of discrepancy in expansion timing between ND4 (10,000-13000 years) and COI (50000 years) and its implications for P. argentipes expansion, and (ii) omitting repeat statemen

Response- We appreciate the Editor’s guidance regarding the need to refine the Discussion section. Accordingly, the Discussion has been revised to (i) address the observed discrepancy in population expansion timing inferred from ND4 (10,000–13,000 years) and COI (approximately 50,000 years), including possible biological and evolutionary explanations, and (ii) remove repetitive statements to ensure clarity and conciseness. These revisions have been incorporated in the updated Discussion section (Lines 55-57 in the revised manuscript).

Reviewer 2-

Comment- The manuscript has improved and is much clearer now.

Response- We sincerely thank the reviewer for their positive feedback and appreciation of the improvements made to the manuscript.

Comment- One element that did come through. Have the authors considered comparing different collection methods? i.e., just cattle baited traps. This may eliminate some differences in species behaviour / sibling species.

Response- We thank the reviewer for this valuable observation. In the present study, four different collection methods cattle-baited traps, manual collections, sticky traps, and light traps were employed across sampling sites to ensure comprehensive representation of P. argentipes s.l. populations. The use of multiple standardized trapping methods helped to minimize potential bias arising from differences in species behavior or sibling species composition. As the primary focus of this study was population genetic structure and demographic history, detailed comparisons among trapping methods were beyond its scope. However, we agree that future investigations focusing on trap efficiency and behavioral variation would further enrich the understanding of P. argentipes ecology in Sri Lanka.

Reviewer 3-

Comment- The authors have revised the manuscript “Demographic history and population structure of Phlebotomus argentipes (Diptera: Psychodidae) complex, the leishmaniasis vector in Sri Lanka” in response to the earlier review round. I acknowledge the substantial effort made to address the prior comments, and many issues have been improved. However, a number of important points still require further attention before the manuscript can be considered ready for publication.

Response- We sincerely thank the reviewer for recognizing our efforts to improve the manuscript and for the constructive feedback provided throughout the review process. We have carefully re-evaluated all remaining comments and have made further revisions to address each point in detail. The revised manuscript has been strengthened accordingly, particularly in the sections related to methodology, data interpretation, and discussion.

Comment- Line 41: The stated aim should be consistent: “to investigate demographic history and population genetic structure.”

Response- Thank you for your comment. The correction has been made in the revised manuscript, line 41.

Comment- The concatenated dataset is reported in the results but not mentioned in the methods description. Please correct this.

Response- Thank you for your comment. The correction has been made in the revised manuscript, lines 45-47.

Comment- The Mantel test showed only weak, non-significant correlations. Marker-wise reporting adds unnecessary detail. A clearer sentence would be:

“Mantel tests showed weak, non-significant correlations between genetic and geographic distance, indicating no evidence of isolation by distance.”

Response- Thank you for your comment. The correction has been made in the revised manuscript, lines 51-53.

Comment- Lines 51–54: The statement is too long and should be simplified for clarity.

Response- Thank you for your comment. The correction has been made in the revised manuscript, lines 57-59.

Comment- Line 54: The focus is only on ND4 results, but a conclusive summary should be drawn from the overall BSP analysis, including both COI and ND4.

Response- Thank you for your comment. The correction has been made in the revised manuscript, lines 55-57.

Comment- Line 55: This sentence appears incomplete and does not fit the context. Please revise.

Response- Thank you for your comment. The revision has been made in the corrected manuscript, lines 57-59.

Comment- While the authors now note that COI and ND4 data were reused from a previous study, the Methods section still describes sample collection and DNA extraction in detail, giving the impression that new experimental work was conducted. This is misleading. Please rewrite this section to explicitly state that no new laboratory work was performed, and that the dataset was reused exclusively for additional analyses.

Response- We appreciate the reviewer’s valuable comment. We have now revised the Materials and Methods section to clearly state that no new laboratory work was conducted for this study. The correction has been mad in lines 115-128.

Comment- Please remove the STRUCTURE analysis. I regret suggesting its inclusion in the previous round. Upon re-evaluation, STRUCTURE is not appropriate for mitochondrial markers (due to their haploid, maternally inherited, non-recombining nature). Its inclusion is methodologically unsound and should be omitted from the manuscript.

Response- We thank the reviewer for the clarification. We fully agree that STRUCTURE analysis is not appropriate for mitochondrial datasets due to their haploid, maternally inherited, and non-recombining nature. In fact, this analysis was not part of our original submission, as we were aware of these methodological constraints and had initially focused on analyses better suited to mitochondrial data (e.g., haplotype-based and population differentiation indices). The STRUCTURE analysis was incorporated only in response to the reviewer’s earlier suggestion during the first revision. As advised now, we have removed this section and all related figures and references from the manuscript.

Comment- Lines 197–198: Table 1 does not present average pairwise differences. Please correct the description.

Response- Thank you for your comment. The correction has been made in the revised manuscript, line 180.

Comment- Table 1 legend: Remove the word “separately.”

Response- Thank you for your comment. The correction has been made in the revised manuscript, lines 187.

Comment- Line 200: Add “respectively” at the end of the sentence.

Response- Thank you for your comment. The correction has been made in the revised manuscript, lines 183.

Comment- Line 217: The Mantel test description highlights only ND4, but similar values were obtained for the concatenated dataset. Please be consistent.

Response- We thank the reviewer for pointing this out. Upon rechecking, we realized that the sign of the correlation coefficient for the concatenated dataset was incorrectly reported. The corrected Mantel test results are: COI, r = –0.038 (p = 0.48); ND4, r = 0.040 (p = 0.52); concatenated, r = –0.043 (p = 0.43). None of the correlations are statistically significant, indicating no isolation by distance among populations. The text has been revised accordingly to reflect these corrected values and to consistently indicate that no dataset shows a significant. The correction has been made in the revised manuscript, lines 209

Comment- Line 270: The Fst values appear to be inappropriately interpreted. Please review carefully.

Response- Thank you for your comment. The correction has been made in the revised manuscript, lines 243.

Comment- Lines 280–284: The description of the mismatch distribution analysis is unclear and should be rewritten for clarity.

Response- Thank you for your comment. The unclear part was rewritten in revised manuscript lines 252-258.

Comment- Lines 285–292: The BSP results should be presented marker by marker (COI and ND4) before providing a combined conclusion. Currently, COI is overlooked.

Response- Thank you for your comment. The correction has been made in the revised manuscript, lines 259-262.

Comment- Paragraph 2: Please expand discussion on the species complex issue and how it may affect the interpretation of results.

Response- Thank you for your comment. The correction has been made in the revised manuscript, lines 295-300.

Comment- Lines 323–325: Fst values are consistently <0.027 across all populations. It is not correct to suggest different dispersal behaviors in certain populations. This statement needs revision.

Response- Thank you for your comment. The correction has been made in the revised manuscript, lines 301-317.

Comment- Lines 323–324: The paragraph contains several general statements, making it confusing. Please rewrite more clearly.

Response- The correction has been done in the revised manuscript.

Comment- Line 332: References appear misplaced; check and correct.

Response- Thank you for your comment. The correction has been made in the revised manuscript, lines 323.

Comment- Line 335: The discussion focuses only on ND4-based expansion timing (~10,000–13,000 years) while COI indicates an older signal (~50,000 years). This discrepancy should be discussed, particularly regarding its implications for P. argentipes expansion.

Response- Thank you for your comment. The correction has been made in the revised manuscript, lines 326- 336.

Comment- Lines 340–342: The sentence is unclear and requires revision.

Response- Thank you for your comment. The correction has been made in the revised manuscript, lines 339-343.

Comment- Lines 343–344: The phrase “possible relation” is vague. Please specify what relationship is being discussed.

Response- Thank you for your comment. The correction has been made in the revised manuscript, lines 342-345.

Comment- The conclusion that ND4 is the most informative marker should be highlighted and discussed in the main text, not only in passing.

Response- Thank you for your comment. The correction has been made in the revised manuscript.

Comment- Overall, the Discussion is too lengthy in places and includes general or repetitive statements. It should be revised to be more concise, focusing on the observations and their implications.

Response- The correction has been done in the revised manuscript.

Comment- Ensure species names are written consistently throughout the text.

Response- We appreciate the reviewer’s comment and have carefully checked the manuscript to ensure consistency in species nomenclature. The term Phlebotomus argentipes s.l. (sensu lato) is used throughout the text to represent the species complex (synonym with P.argentipes complex), within which sibling species coexist sympatrically. At its first appearance, the full name Phlebotomus argentipes is provided, and the abbreviated form (P. argentipes s.l.) is used thereafter in accordance with taxonomic conventions. The same format is consistently applied across text, tables, and figures to maintain clarity and scientific accuracy.

Comment- Expand abbreviations only once, at first mention.

Response- We appreciate the reviewer’s observation. All abbreviations have now been expanded only once at their first mention in the manuscript, following the journal’s style requirement. Previously, abbreviations were defined at the first appearance of each section, However, in accordance with formal writing conventions, full terms were retained at the beginning of paragraphs or sentences to maintain readability and grammatical correctness.

---

## [Decision Letter · Decision Letter 2]

9 Nov 2025

Demographic history and population structure of Phlebotomus argentipes (Diptera: Psychodidae) complex, the leishmaniasis vector in Sri Lanka

PONE-D-25-27639R2

Dear Dr. De Silva,

We’re pleased to inform you that your manuscript has been judged scientifically suitable for publication and will be formally accepted for publication once it meets all outstanding technical requirements.

Kind regards,

Swaminathan Subramanian, Ph.D.

Academic Editor

PLOS ONE

Additional Editor Comments (optional):

Reviewers' comments:

Reviewer's Responses to Questions

**Comments to the Author**

Reviewer #2: All comments have been addressed

Reviewer #3: All comments have been addressed

2. Is the manuscript technically sound, and do the data support the conclusions?

Reviewer #2: Yes

Reviewer #3: Yes

3. Has the statistical analysis been performed appropriately and rigorously?

Reviewer #2: Yes

Reviewer #3: Yes

4. Have the authors made all data underlying the findings in their manuscript fully available?

Reviewer #2: Yes

Reviewer #3: Yes

5. Is the manuscript presented in an intelligible fashion and written in standard English?

Reviewer #2: Yes

Reviewer #3: Yes

Reviewer #2: While the paper has attempted to address the question of why species identification was not carried out, this should be considered a weakness of the study and highlighted as such. The authors are correct this is maternally inherited mitochondrial DNA, however, the sibling species are, as they state, reproductively isolated.

Reviewer #3: (No Response)

**Do you want your identity to be public for this peer review?** For information about this choice, including consent withdrawal, please see our Privacy Policy

Reviewer #2: **Yes: ** Michael Coleman

Reviewer #3: No

---

## [Editor Report · Acceptance letter]

PONE-D-25-27639R2

PLOS ONE

Dear Dr. De Silva,

I'm pleased to inform you that your manuscript has been deemed suitable for publication in PLOS ONE. Congratulations! Your manuscript is now being handed over to our production team.

Kind regards,

on behalf of

Dr. Swaminathan Subramanian

Academic Editor

PLOS ONE